# Sorting Out Quantum Monte Carlo

## Abstract

Molecular modeling at the quantum level requires choosing a parameterization of the wavefunction that both respects the required particle symmetries, and is scalable to systems of many particles. For the simulation of fermions, valid parameterizations must be antisymmetric with respect to the exchange of particles. Typically, antisymmetry is enforced by leveraging the anti-symmetry of determinants with respect to the exchange of matrix rows, but this involves computing a full determinant each time the wavefunction is evaluated. Instead, we introduce a new antisymmetrization layer derived from sorting, the *sortlet*, which scales as $O(N \log N)$ with regards to the number of particles – in contrast to $O(N^3)$ for the determinant. We show numerically that applying this anti-symmeterization layer on top of an attention based neural-network backbone yields a flexible wavefunction parameterization capable of reaching chemical accuracy when approximating the ground state of first-row atoms and small molecules.

## 1 Introduction

### 1.1 Overview

Quantum Monte Carlo (QMC) methods are a class of algorithm that aim to model the wavefunction for a system of quantum particles, typically an atom or molecule. In its simplest form, Variational Quantum Monte Carlo (VQMC) is a QMC method that aims to estimate the lowest-energy state of a system via the variational principle. This is done by minimizing the Rayleigh quotient of the system's Hamiltonian over a parametric family of wavefunctions – commonly referred to as the *wavefunction ansatz*. Variational principles from functional analysis yield that under mild assumptions, the minimizer approaches the true ground state wavefunction as the parametric family increases in expressiveness.

The first method resembling modern VQMC was proposed by London and Heilter in the late 1920s [9], where they attempted to calculate the ground state of the diatomic hydrogen molecule. Fermi and then later Kalos [21] converted the problem into a Monte Carlo Sampling one. The advent of the practical Slater determinant ansatz in the 1950s and the growth in available computational power since has allowed QMC algorithms to become one of the benchmark frameworks for deriving the properties of chemical systems *in silico*. Often, QMC is used to benchmark other non-sampling methods, such as Coupled Cluster methods [2] and Density Functional Theory (DFT).

Advances in automatic differentiation complemented by empirical ML experience have recently produced new types of wavefunction ansatz, assembled around deep neural network backbones. Neural networks are particularly attractive in this setting, due to their favourable scalability as dimension increases. Electron configuration space, $\mathbb{R}^{3N}$, grows exponentially in the number of electrons $N$, rendering this ability critical. Hybrid methods, seeking to combine Slater determinants with neural orbitals or Jastrow factors, have recently shown promise on systems comprised of a large number of particles – even when relatively few Slater determinants are employed [34]. Their performance is especially notable when contrasted against existing Hartree-Fock implementations with an equivalent number of determinants.

Despite these recent successes, there is a good reason to look beyond Slater determinants when designing ansatz – determinants are relatively computationally expensive. Evaluating a

determinant-based ansatz scales $O(N^3)$ in the number of electrons, devolving to $O(N^4)$ when evaluating the local energy (due to the Laplacian). Estimating the energy is required at every step of nearly every QMC algorithm (variational or otherwise), so this quickly becomes a bottleneck if we look to scale QMC beyond small molecules to problems of practical interest, where $N$ could be on the order of thousands or even tens of thousands of electrons.

## 1.2 Our Contribution

Seeking to reduce this complexity, we introduce a novel antisymmetrization operation, the *sortlet*, and apply it within the VQMC framework to model ground states of various atoms and molecules. Crucially, our operation enjoys an improved $O(N \log N)$ complexity, which comes from replacing the determinant with a cheaper alternative: sorting. Sorting is known to be universal for systems of 1d electrons [40; 20], and has been alluded to previously as a method for designing ansatz (see Appendix B of [34] and [24]). Our contribution is twofold – we show both that a natural extension of sorting for electrons in 3d space exists, as well as that this ansatz can achieve chemical accuracy, at least on small systems, when paired with a sufficiently flexible functional backbone such as a modern attention based neural network.

Prior work [31; 1] explored the usage of Vandermonde determinants which scale $O(N^2)$, but both were unable to achieve the high degree of accuracy required for applications in quantum chemistry. In Section 5.2 we present a topological obstruction to a single continuous Vandermonde determinant learning the exact ground state, based on the known geometry of the wavefunction's nodal surface.

The construction of a universal, sub-cubic-time continuous representation of ground-state wavefunctions remains an open problem. However, we show that, like the vandermonde determinant, our sortlet can represent the ground state exactly if we allow discontinuous parameterizations. We also highlight a key benefit of the lower scaling complexity of the sortlet – allowing more terms in the wavefunction ansatz with the same (asymptotic) computational budget. Classical QMC techniques have relied on

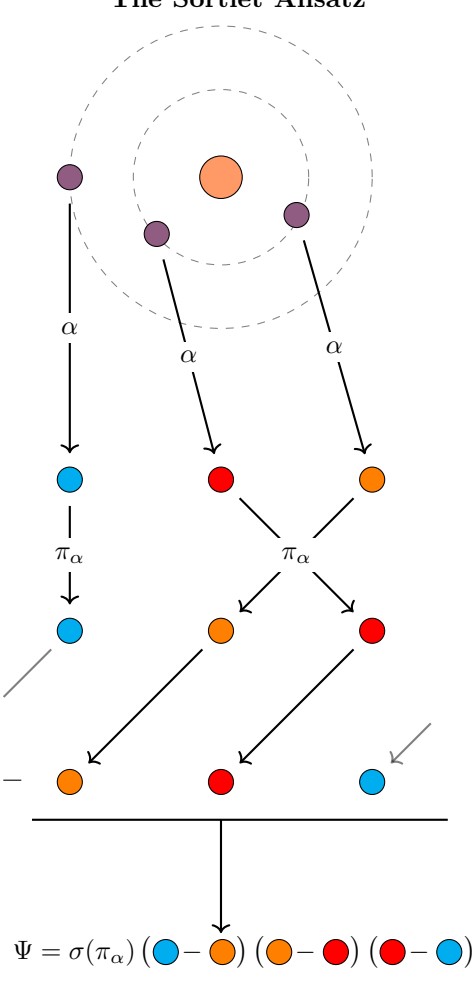

**The Sortlet Ansatz**

$$\Psi = \sigma(\pi_\alpha) \left( \bullet - \bullet \right) \left( \bullet - \bullet \right) \left( \bullet - \bullet \right)$$

Figure 1: Geometric illustration of the Sortlet ansatz construction given in Equation (5). Here $\pi_\alpha$ is the permutation that sorts the output of $\alpha(r)$.

larger expansions to help mitigate topological inconsistencies between the parametric ansatz and the true ground state [3; 5]. In the context of VQMC, this might partially explain why our method is able to achieve higher accuracy than previous Vandermonde constructions [31; 1].

Numerically, we demonstrate that combining our sortlet antisymmetrization layer with the PsiFormer [41] attention backbone is sufficient to achieve chemical accuracy on a handful of atoms and small molecules, as well as reproduce the potential energy surface of $H_4$ rectangles as one bond length is scanned. While wavefunction ansatz built from alternatives

to determinants have existed in the QMC literature for some time, to the best of our knowledge, this is the first work to demonstrate chemical accuracy with something other than a determinant – even if only on small molecular systems at the current stage. Flexibility of the attention neural network backbone of [41] offers another partial explanation for why our ansatz has proven more accurate than other full determinant alternatives in the past. That said, at the current state our results are far from competitive with those of neural network ansatz with full determinants [34; 38; 41]. In fairness to our approach, the results presented in that body of work are the outcome of a long series of incremental improvements – far beyond the scope of the initial proof-of-concept described in this paper. Our accuracy on small systems, we believe, is a evidence that the sortlet ansatz is a promising direction for further study, and that with similar investments in software engineering, our method could become competitive on more difficult benchmarks.

### 1.3 Where did the determinants come from anyway?

Determinants appear in nearly all QMC ansatz as a simple mathematical way to satisfy the generalized Pauli Exclusion Principle: any valid wavefunction must be antisymmetric under exchange (transposition) of any two electrons with the same spin. Intuitively, this follows from the idea that quantum particles such as electrons are indistinguishable, so changing their order (which has no physical meaning) cannot change the state of the system. Explaining the appearance of the -1 factor is more involved – representing the fundamental difference between Fermions and Bosons (see [17]) – but for our purposes, we just assume any valid wavefunction for Fermions needs to respect the antisymmetry equation:

$$\Psi(r_1^\uparrow, \cdots, r_M^\uparrow, r_1^\downarrow, \cdots, r_i^\downarrow, \cdots, r_j^\downarrow, \cdots) = -\Psi(r_1^\uparrow, \cdots, r_M^\uparrow, r_1^\downarrow, \cdots, r_j^\downarrow, \cdots, r_i^\downarrow, \cdots) \quad \text{(AS)}$$

Alternation of the determinant under the exchange of any two rows or columns has been conventionally employed to satisfy this property. Originally, the Slater determinants typically consisted of parameterizing a matrix function $\Phi : \mathbb{R}^{3N} \longrightarrow \mathbb{R}^{N \times N}$ via a collection of $N$ single electron orbitals $\Phi_{ij} = \phi_i(r_j)$, each of which was only a function of a single $r_j$. Upon interchanging the electron positions, the resulting matrix $\Phi'_{ij}$ is exactly $\Phi_{ij}$ with two rows swapped, flipping the sign of $\Psi := \det[\Phi_{ij}]$.

Though this approach is undeniably tidy, it suffers limited expressive power due to its inability to accurately model electronic correlation, which involves the interactions of all N-particles at a time rather than treating particles as mean-field orbitals. Common remedies include either the addition of parametric Jastrow factors which are multiplicative against the determinant, i.e $\Psi = e^{J(r)} \det[\Phi_{ij}(r)]$, or backflow transformations [10], which makes each orbital $\phi_i(q_j)$ dependent on all electron positions through so called pseudo-coordinates $q_j = r_j + \varepsilon^i r_i$. Modern neural network approaches such as FermiNet, PsiFormer or PauliNet can be seen as a more flexible generalization of backflow, since they opt to parameterize the matrix $\Phi_{ij}$ as the output of a deep neural network, but in a way that the $\Phi_{ij}$ depend symmetrically on all electron positions except for $r_j$, which is allowed non-symmetric influence. Filling out $\Phi_{ij}$ this way preserves the antisymmetry of the Slater determinant, and provides sufficient flexibility to capture electronic correlation. Large neural networks and powerful parallel accelerators have allowed this approach to scale to systems of around 40 electrons with state-of-the-art results [41].

## 2 Variational Quantum Monte Carlo

The following is a brief summary of the core aspects VQMC, but it is by no means exhaustive. Those experienced with the framework will likely find it redundant, and those completely new will probably find it incomplete. For the latter group, we recommend either [4] or [25], which provide much more exhaustive treatments of the relevant background.

### 2.1 The Born-Oppenheimer Hamiltonian

While our wavefunction ansatz is general enough to describe any system of Fermions, in this paper we will focus on quantum chemical systems comprised of electrons. Similar to

[34; 38] we will work in the Born-Oppenheimer approximation, where the molecular nuclei are treated as fixed, and the Hamiltonian governing the forces acting on the electrons is given by

$$\hat{H} = -\frac{\nabla^2}{2} + \left[ \sum_{i>j} \frac{1}{|r_i - r_j|} - \sum_{i,I} \frac{Z_I}{|r_i - R_I|} + \sum_{I>J} \frac{Z_I Z_J}{|R_I - R_J|} \right] \quad \boxed{\begin{array}{l} r_i: \text{ electron positions} \\ R_i: \text{ nuclei positions} \\ Z_I: \text{ nuclear charges} \end{array}}$$

(BO)

The terms inside the square brackets are often written simply as $V(r)$, since they represent the potential energy of the electrons. Analogously, $-\nabla^2$ is the kinetic energy operator, and interpreted as the squared momentum operator $p = i\nabla$, since $\nabla^2 = \text{div}(\nabla) = \langle i\nabla, i\nabla \rangle$. The ground state $\Psi_g$ for a given system $\hat{H}$, is nothing more than the eigenfunction corresponding to the smallest eigenvalue $E_1$ of $\hat{H}$.

## 2.2 THE VARIATIONAL PRINCIPLE

The *variational* aspect of VQMC comes from the following proposition:

**Proposition 1.** *Variational Principle*
*Let $\Psi_1$ be minimum eigenvector of the Hermitian operator $\hat{H}$, that is, the eigenfunction associated to the smallest eigenvalue $E_1$. Then*

$$\Psi_1 = \underset{\Psi}{\text{argmin}} \frac{\left\langle \Psi | \hat{H} | \Psi \right\rangle}{\langle \Psi | \Psi \rangle}$$

(VAR)

The term inside argmin is often called the *Rayleigh quotient* and so denoted by $R(\Psi)$. Proving this proposition amounts to hardly more than a little bookkeeping, see Appendix A.1 The variational principle comes in very handy, providing a straightforward technique to solve for the ground state $\Psi_1$ – simply minimize the Rayleigh $R$ quotient over all admissible $\Psi$. Practically, however, there are a few complications, which we'll discuss next.

## 2.3 THE LOCAL ENERGY AND ITS VANISHING VARIANCE

The variational energy $E = R(\Psi_\theta)$, for any parameteric configuration of $\theta$ can in principle be computed from the inner product in (VAR). Since we are limited to approximating the energy with a finite average in practice, this is suboptimal – the variance will be prohibitive, and we will need a tremendous number of samples to achieve a reasonable estimate. A much better approach is to instead define the local energy (at any point $x \in \mathbb{R}^{3N}$ )

$$E_{\text{loc}}(\Psi_\theta)(x) = \frac{[\hat{H}\Psi_\theta](x)}{\Psi_\theta(x)}$$

(1)

it is easy to then see that

$$\mathbb{E}_{x \sim \Psi_\theta^2}[E_{\text{loc}}(x)] = E$$

where $x \sim \Psi_\theta^2$ is taken to mean that the $x$ are drawn from the normalized distribution, $\frac{\Psi_\theta^2}{\langle \Psi_\theta | \Psi_\theta \rangle}$. Reformulating the energy approximation in terms of $E_{\text{loc}}$ instead of the Rayleigh quotient has two immediate advantages

1. Since we are sampling from $\Psi_\theta^2$, we avoid regions of low importance, significantly reducing the variance of our estimator.
2. The local energy is invariant to rescaling by any constant, so normalizing the wavefunction $\Psi_\theta$ is unnecessary. Combined with the ability of MCMC to (asymptotically) draw samples from the un-normalized density $\Psi_\theta^2$, we can avoid estimating the partition constant altogether.

VQMC also benefits from a third seemingly magical property: as we update $\theta$ to minimize $E$, the variance *vanishes*. To see why, consider the following:

**Proposition 1.** *Let $\Psi_1$ be the eigenfunction associated to the ground state energy $E_1$ (as before), then*

$$E_{loc}(\Psi_1)(x) = E_1 \qquad \forall x \in \mathbb{R}^{3N} \tag{2}$$

*i.e the local energy is constant at optimality. Assuming that $\Psi_\theta \longrightarrow \Psi_1$ as we optimize $\theta$, the variance of (1) decreases, meaning our estimates of the energy and its gradients improve as $\Psi$ approaches the ground state.*

*Proof.* Equation (2) is a direct consequence of the variational principle; $\Psi_1$ is an eigenvector of $\hat{H}$, thus $\hat{H}\Psi_1 = E_1\Psi_1$, which yields

$$E_{\text{loc}}(\Psi_1)(x) = \frac{\hat{H}\Psi_1(x)}{\Psi_1(x)} = \frac{E_1\Psi_1(x)}{\Psi_1(x)} = E_1$$

$\square$

### 2.4 Derivatives of the Energy Functional

Performing any sort of first order optimization over the parameters of our ansatz $\Psi_\theta$ to minimize its energy, denoted $E(\theta)$, will also require its gradients. Following the preceding section, we will estimate $E(\theta)$ via the following Monte-Carlo estimate

$$E(\theta) \approx \bar{E}(\theta) = \sum_{i=1}^{N} E_{\text{loc}}(\Psi_\theta)(x_i) \qquad x_i \sim \Psi_\theta^2 \tag{3}$$

Any MCMC method (such as Random-Walk Metropolis Hastings for example) can be used to draw the $x_i$, which we will detail further in the next section. Approximating the energy via (3), while superior for the reasons laid out above, comes at the price of making the computation of its gradients more involved than naively applying automatic differentiation to each term in the finite sum. Differentiating $\frac{\partial E}{\partial \theta}$ correctly requires also accounting for the influence of the parameters on the distribution $\Psi_\theta^2$ which the $x_i$ are drawn from. Like in many other other works [34; 41; 4; 1; 38; 13], we use the gradient estimator

$$\nabla E(\theta) = 2\frac{1}{n}\sum_{i=1}^{n}\left[ E_{\text{loc}}(x_i)\nabla \log \Psi_\theta(x_i) - 2\left(\frac{1}{n}\sum_{i=1}^{n}[E_{\text{loc}}(x_i)]\right)\nabla \log \Psi_\theta(x_i)\right] \tag{4}$$

where $x_i \sim \Psi_\theta^2$ are drawn using MHMC, but we offer a full derivation in Appendix A.2. Notably, this can be seen as an instance of the REINFORCE estimator widely used in reinforcement learning [16].

## 3 The Sortlet Ansatz

### 3.1 The Sortlet Ansatz

We are ready to define what we coin the *Sortlet Ansatz*. First, let's define what we call a *sortlet*, denoted by $\Psi_\alpha$:

$$\Psi_\alpha(r) = \sigma(\pi_\alpha)\prod_{i=1}^{N}(\alpha_{i+1}(r) - \alpha_i(r)) \tag{5}$$

where

- $\alpha : \mathbb{R}^{N\times 3} \longrightarrow \mathbb{R}^N$, permutation equivariant, and reindexed such that $\alpha_i < \alpha_{i+1}$ : $1 \le i \le N-1$, and $\alpha_{N+1} = \alpha_1$. **This is the neural network backbone.**

- $\pi_\alpha$ is the permutation on $N$ letters which re-indexes (sorts) the tuple $\alpha(r)$

- $\sigma(\pi_\alpha)$ is the parity, equivalently number of transpositions in $\pi_\alpha$, mod 2.

Analogously to other VQMC ansatz, we will employ an expansion of sortlets to serve as our sortlet ansatz in VQMC. Specifically, our $\Psi$ is expanded as

$$\Psi = \exp[-J_\beta] \sum_{i=1}^{K} \left[ \Psi_{\alpha_i} \sum_{I} e^{-\gamma_I \sum_j |r_j - R_I|} \right] \tag{6}$$

where just as in [38; 41], (with $\rho_i$ returning the spin of electron $i$) the simple Jastrow factor that we decided to employ is:

$$J_\beta = \sum_{i<j,\rho_i=\rho_j} \frac{1}{4} \frac{\beta_1}{\beta_1^2 + |r_i - r_j|} \sum_{i<j,\rho_i\neq\rho_j} \frac{1}{2} \frac{\beta_2}{\beta_2^2 + |r_i - r_j|} \tag{7}$$

As we will show in Section 3.2, with discontinuous $\alpha$ we can exactly represent the ground state wavefunction with a single sortlet.

**Complexity:** The sortlet for a given $\alpha$ can be computed in time $O(N \log N)$. Taking $K = O(N)$ and with the assumption $\alpha(r)$ can be evaluated in time $O(N^2)$, we have that $\Psi$ can be evaluated in time $O(N^2 \log N)$, significantly faster than a determinant.

## 3.2 Properties of the Sortlet Ansatz

**Proposition 2.** *The Sortlet Ansatz satisfies the generalized Pauli exclusion principle – it is anti-symmetric to the exchange of electrons with the same spin.*

*Proof.* The product $\prod_{i=1}^{N}[\alpha_{i+1} - \alpha_i]$ is invariant by design since the terms are always sorted before taking the pairwise difference, negating any permutation. $\sigma(\pi_\alpha)$ flips sign upon each odd permutation, since $\sigma$ is multiplicative, and undoing the permutation then sorting the original array is equivalent to sorting the permutated array, i.e if $\tau$ is any transposition of two electrons with the same spin, $\pi_{\alpha(r)}\tau^{-1} = \pi_{\alpha(\tau r)}$ and so

$$\sigma(\pi_{\alpha(\tau r)}) = \sigma(\pi_\alpha)\sigma(\tau^{-1}) = -\sigma(\pi_\alpha) \tag{8}$$

since any transposition has parity $-1$. $\square$

Note that to avoid over-constraining the ansatz by also making it anti-symmetric to transposition of opposite spin electrons – something the true wavefunction is **not** required to satisfy – we use the same trick as in [41], attaching a spin $\pm 1$ term to each input electron coordinate $r_i$ as $(r_i, \pm 1)$, which breaks equivariance of the output $\alpha(r)$ for those pairs.

| Molecule | Exp. Energy | $O(N^2)$ Ansatz | | $O(N^3)$ Ansatz | |
| | | Sortlet (Ours) | Vandermonde [31] | Non-learned QMC[33; 39] | FermiNet [34] |
|---|---|---|---|---|---|
| Li | -7.4780 | -7.477(8) | -7.4782 | -7.4780 | -7.4779 |
| LiH | -8.0705 | -8.070(3) | - | -8.070 | -8.7050 |
| Be | -14.6673 | -14.667(1) | -14.6673 | -14.6671 | -14.6673 |
| Li2 | -14.9947 | -14.994(5) | - | -14.9555 | -14.9947 |
| B | -24.6539 | -24.652(7) | 24.5602 | -24.6533 | -24.6537 |
| C | -37.8450 | -37.83(1) | -37.3531 | -37.8437 | -37.8447 |
| N | -54.5892 | -54.01(8) | -53.1855 | -54.5873 | -54.5888 |
| CH4 | - | -40.20(1) | - | -40.4416 | -40.5140 |

Figure 2: Results from applying the Sortlet ansatz to a selection of atoms and small molecules. Energy values are all given in Hartree (Ha) atomic units. Chemical accuracy is defined to be within 1.5 mHa of the experimental values, green denotes values within this tolerance, red outside. Uncertainty is included in the parenthesis around the last digit – see Appendix B.

**Proposition 3.** *Each $\Psi_\alpha$ is continuously once-differentiable on $\mathbb{R}^{3N}$, meaning $\frac{\partial \Psi_\alpha}{\partial r^i}$ is continuous for all $i$.*

We defer the full proof to Appendix A.3.

## 4 NUMERICAL EXPERIMENTS

### 4.1 EVALUATION ON FIRST ROW ATOMS AND SMALL MOLECULES

Section 3.1 collects the results of running QMC with our Sortlet ansatz on the ground state for a small collection of first row atoms and small molecules. We see that up to and including Boron, we are able to recover the ground state experimental energy to chemical accuracy. For Carbon and beyond, while we no longer achieve chemical accuracy, our method still outperforms the previous $O(N^2)$ method of [31].

### 4.2 PERFORMANCE AS A FUNCTION OF SORTLETS ON BORON

In Figure 3a, we show an ablation study comparing performance in terms of absolute difference to the empirical ground state values after $50,000$ optimization steps, against the number of sortlets on Boron. While all except one of the runs terminate at less than 20 mHa to the ground state, we see that those with $K > 16$ reach a slightly lower error on average, but achieve that value significantly faster than those with $K < 16$. Size of the network is kept constant for all runs (with the exception of the linear output layer), the only variable explored in this study is the number of terms $K$.

### 4.3 REPRODUCTION OF $H_4$ POTENTIAL ENERGY CURVE

In Figure 3b we reproduced the $H_4$ experiment from [34], where the geometry of the $H_4$ rectangle was linearly adjusted by varying the bond length of the bottom two hydrogen atoms. In agreement with FermiNet, we were able to reproduce the curve showing a maximum at $\Theta = 90 \deg$, not a minimum like the benchmark Coupled Cluster calculations.

### 4.4 DIRECT COMPARISON TO VANDERMONDE ANSATZ

In Section 4.4 we directly compare training curves of vandermonde ansatz vs those of the Sortlet ansatz *with the same number of terms in their expansion*. We see that even with effectively more computational resources (since each term is $O(N^2)$), the Vandermonde ansatz is still unable to outperform the Sortlet.

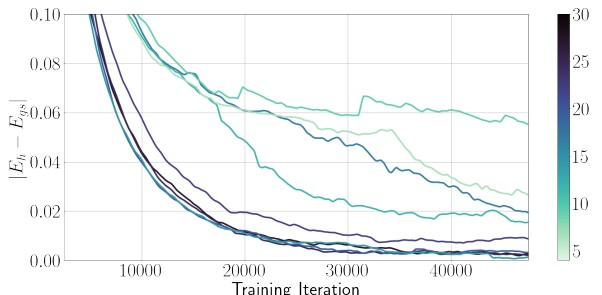
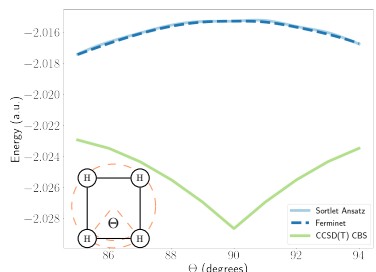

(a) Comparing the number of Sortlets $K$ (color) against the error to the empirical ground state energy.

(b) Reproduction of $H_4$ potential energy surface from the FermiNet paper.

Figure 3: Boron Sortlet Ablation and $H_4$ rectangle.

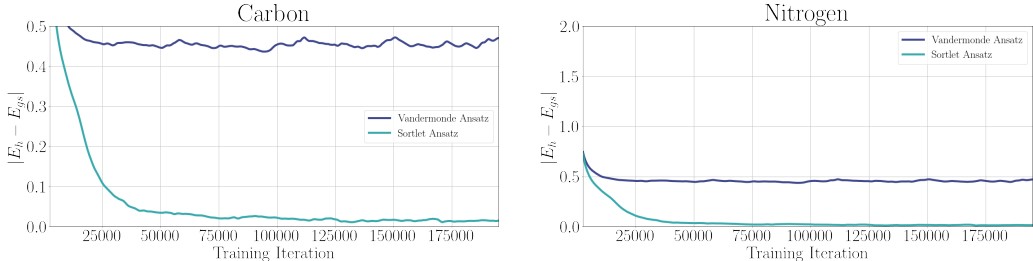

Figure 4: Comparison of Vandermonde and Sortlet Ansatz

## 5 Comparison with other Pairwise Constructions and Universality

In this section, we show a weak form of universality of the sortlet and that of the Vandermonde construction put forward in [1; 31]. We show that a slight modification of Theorem 2 in [31] also yields universality for the sortlet, but similar to the argument in the Appendix of [34], requires discontinuous functions. Theoretical universality so remains an open question, but we finish by discussing the implications of the nodal structure on experimental results.

### 5.1 Non-smooth universality

A nearly identical construction to Theorem 2 in [31] recovers a similar type of universality for the sortlet:

**Proposition 4.** *For a given ground state wavefunction $\Psi_g$, if we allow $\alpha$ to be discontinuous, setting*

$$\alpha_j = \pi^*(r, j) \left( \frac{\Psi^g \left[ \pi^*(r)r \right]}{(N-1)} \right)^{1/N} \tag{9}$$

*yields that $\Psi_\alpha = \Psi_g$. Here $\pi^*(r)$ is the permutation on $N$ letters that maps $r \in \mathbb{R}^{3N}$ back to the fundamental domain (see [7]), $\pi^*(r, j) \in [1, N]$ is the index the $j$th electron maps to.*

*Proof.* The proof is immediate from the definition of $\Psi_\alpha$ in Equation (5)

$$\Psi_\alpha = \sigma(\pi^*) \prod_{i=1}^{N} [\alpha_{i+1} - \alpha_i] = \sigma(\pi^*) \left( \prod_{i=1}^{N-1} \left[ \frac{|\Psi_g|}{(N-1)} \right]^{1/n} \right) (N-1) \left[ \frac{|\Psi_g|}{(N-1)} \right]^{1/n} = \sigma(\pi^*)|\Psi_g| \tag{10}$$

$\square$

### 5.2 Comparison with the Vandermonde Ansatz

In [18; 1] and later [31], a similar type of pairwise antisymmetric ansatz was proposed, which takes the form

$$\psi'_{\text{pair}}(r) = \prod_{i<j} [\phi_B(r_i, \{r\}) - \phi_B(r_j, \{r\})] \tag{11}$$

where the second argument to $\phi_B$ is denoted as such to indicate the function is invariant to permutations of the electron positions, except for $r_i$ (as our $\alpha$ has been assumed all along). This expression is also equivalent to a Vandermonde matrix determinant (see Equation 1 [31]), which is why we refer to it simply as the Vandermonde Ansatz. Pang et al claim to have proved this form is universal for ground states, but unfortunately a simple argument shows this is not quite accurate. Ground states for small atoms (Li, Be) have been proven to possess exactly two nodal domains [26] – open sets $\mathcal{N} \subseteq \mathbb{R}^{3N}$ where the wavefunction is non-zero. So to represent them exactly, the ansatz must also have two nodal domains. Below we prove that for $\psi'_{\text{pair}}$ this is not the case – regardless of the parameterization of $\phi_B$, $\psi'_{\text{pair}}$ always has at least 4 nodal domains.

**Proposition 5.** $\psi'_{pair}$, *as defined in* (11) *has* $N^\uparrow! + N^\downarrow!$, *nodal domains. The ground state is proven to have only 2 for Li, Be (see [5]) so this implies one Vandermonde determinant is insufficient to accurately represent the ground state.*

*Proof.* Let $r \in \mathbb{R}^{3N}$ be an arbitrary configuration, and let $\pi \in S(N^\uparrow) \cup S(N^\downarrow)$ be a permutation on either the spin up or spin down electrons. We claim then that $r$ and $\pi r$ must always lie in separate nodal domains. To see this, let $(i, j)$ be any indices of two electrons with the same spin who are transposed by $\pi$. Note that the product comprising $\psi'_{\text{pair}}$ contains a pairwise term $[\phi_B(r_i, \{r\}) - \phi_B(r_j, \{r\})]$. Without loss of generality, asume $\phi_B(r_i, \{r\}) > \phi_B(r_j, \{r\})$. But for $\hat{r} = \pi r$ we must have

$$[\phi_B(r_i, \{r\}) - \phi_B(r_j, \{r\})] = -[\phi_B(\hat{r}_i, \{\hat{r}\}) - \phi_B(\hat{r}_j, \{\hat{r}\})] \tag{12}$$

(because $\hat{r}_i = r_j$ and $\hat{r}_j = r_i$, and $\phi_B$ is otherwise invariant to $\pi r$). Applying the intermediate value theorem (since $\phi_B$ is assumed continuous), it then follows for any path

$$\gamma(t) : [0, 1] \longrightarrow \mathbb{R}^{3N} \qquad \gamma(0) = r \qquad \gamma(1) = \hat{r} \tag{13}$$

connecting $r, \hat{r}$ must satisfy $[\phi_B(\gamma(t')_i, \{\gamma(t')\}) - \phi_B(\gamma(t')_j, \{\gamma(t')\})] = 0$ for some $t' \in (0, 1)$. Since $\psi'_{\text{pair}} = \prod_{i<j} [\phi_B(r_i, \{r\}) - \phi_B(r_j, \{r\})]$, zero in any $\phi_B(r_i, \{r\}) - \phi_B(r_j, \{r\})$ zeros $\Psi$. Since $\pi \in S(N^\uparrow) \cup S(N^\downarrow)$ was arbitrary, so there are exactly $N^\uparrow! + N^\downarrow!$ nodal domains.

$\square$

**Remark:** Since $\pi$ was arbitrary, this also prohibits the sort of triple exchanges which in the exact ground state, connect same sign regions and collapse the number of nodal cells to 2 [27]. Additionally, although Beryllium is the largest system for which the two nodal domain conjecture is proven, it's frequently conjectured through the literature to be true universally, again see [7; 27; 5]. It's also known that $\pi^*$ is not constant across nodal domains as claimed in [31] – regions were $\pi^*$ is constant are referred to as permutation domains, and each nodal domain contains multiple permutation cells (see Figure 3 in [15]).

## 5.3 Advantages of the Sortlet Ansatz against the Vandermonde Ansatz

To be clear, a single sortlet suffers from the same issue of too many nodal domains proven for the Vandermonde determinant in Proposition 5. Nonetheless, a key advantage of the Sortlet is that, due to the $O(N \log N)$ scaling of each term, we can use an expansion with a linear number of sortlets, $K = O(N)$, while retaining log-quadratic complexity $O(N^2 \log N)$. To achieve the same scaling, a Vandermonde ansatz, would be limited to a logarithmic number of terms. What does this mean practically? Increasing the number of terms is known to be crucial in classical QMC methods, where thousands or even millions of terms can be employed to refine the energy [3]. For Hartree-Fock wavefunctions, increasing the number of terms is essential to reducing the number of nodal domains [27; 26; 5].

Learned neural QMC ansatz such as [34; 38] have mostly bucked this trend, opting to fix the number of terms to a relatively small constant ($K \approx 16$), but increase network size instead. While we are not able to match the flexibility of the determinant PsiFormer, which is likely universal with a constant number of determinants, as seen in Figure 3a, mitigating the nodal inaccuracy of our sortlet by increasing $K$ does seem to increase the speed of convergence.

## 6 Related Work

Neural Networks as representations for wavefunctions were applied to discrete quantum systems in [6; 8; 28; 30; 24] and to continuous bosonic systems in [35]. However, the first models considering fermionic continuous-space systems were DeepWF [18], Ferminet [34], and Paulinet [19]. These approaches were furthered with the introduction of the Psiformer model [42] and works that pushed the framework forward by fine-tuning the techniques and devising new specialized architectures [13; 22; 23; 32] and training procedures [29]. Other directions tried to construct architectures that are easy to sample [40; 43] by leveraging normalizing flows, and meta-learning approaches where the wavefunction is conditioned on the Hamiltonian to transfer to different systems [12; 36; 37; 11]. Sorting as a method of guaranteeing antisymmetry was recently proposed in [40], but limited to the setting where electron positions are restricted to one-dimension.

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
