## A Assorted Proofs

### A.1 Proof of the Variational Principle

*Proof.* Since multiplying $\Psi$ by any complex constant $c \in \mathbb{C}$ leaves the value of $R(\Psi)$ unchanged, we can, without loss of generality, assume that $\langle \Psi | \Psi \rangle = 1$. Taking the orthonormal basis $\Psi_n$ from the application of the spectral theorem to $\hat{H}$, we can expand

$$\Psi = \sum_{n=0}^{\infty} \alpha_n \Psi_n$$

but since $\langle \Psi_i | \Psi_j \rangle = \delta_{ij}$ (1 if $i = j$, 0 otherwise), plugging this expansion into $R$ yields that

$$
\begin{aligned}
R(\Psi) = R\left( \sum_{n=0}^{\infty} \alpha_n \Psi_n \right) \\
= \left\langle \sum_{n=0}^{\infty} \alpha_n \Psi_n \left| \hat{H} \right| \sum_{m=0}^{\infty} \alpha_m \Psi_m \right\rangle \\
= \sum_{n=0}^{\infty} \sum_{m=0}^{\infty} \alpha_n \alpha_m^* \left\langle \Psi_n \left| \hat{H} \right| \Psi_m \right\rangle \\
= \sum_{n=0}^{\infty} \sum_{m=0}^{\infty} \alpha_n \alpha_m^* E_m \underbrace{\langle \Psi_n | \Psi_m \rangle}_{\delta_{mn}} = \sum_{n=0}^{\infty} \alpha_n^2 E_n
\end{aligned}
$$

Since we assumed $\langle \Psi | \Psi \rangle = 1$, it must hold $\sum_n \alpha_n^2 = 1$. This yields that the minimum of $R$ is achieved precisely when

$$\alpha_0 = 1, \alpha_n = 0 : \forall n \geq 1$$

which implies $\Psi = \Psi_0$. $\qquad \square$

### A.2 Derivation of the energy gradient

First, let's rewrite the expectation of $E_{\text{loc}}$ in bra-ket notation

$$\mathbb{E}_{x \sim \Psi_\theta^2}[E_{\text{loc}}(x)] = \frac{\langle \Psi_\theta | E_{\text{loc}} | \Psi_\theta \rangle}{\langle \Psi_\theta | \Psi_\theta \rangle} \tag{14}$$

Differentiating by $\theta$ yields

$$\frac{\partial}{\partial \theta}\left[ \frac{\langle \Psi_\theta | E_{\text{loc}} | \Psi_\theta \rangle}{\langle \Psi_\theta | \Psi_\theta \rangle} \right] = \frac{\partial \langle \Psi_\theta | E_{\text{loc}} | \Psi_\theta \rangle}{\partial \theta} \frac{1}{\langle \Psi_\theta | \Psi_\theta \rangle} + \langle \Psi_\theta | E_{\text{loc}} | \Psi_\theta \rangle \frac{\partial}{\partial \theta} \frac{1}{\langle \Psi_\theta | \Psi_\theta \rangle} \tag{15}$$

For the derivative in the first term, since $\hat{H}$ is Hermitian, we have:

$$\frac{\partial \langle \Psi_\theta | E_{\text{loc}} | \Psi_\theta \rangle}{\partial \theta} = \frac{\partial \left\langle \Psi_\theta | \hat{H} | \Psi_\theta \right\rangle}{\partial \theta} \tag{16}$$

$$= 2 \left\langle \Psi_\theta | \hat{H} | \frac{\partial \Psi_\theta}{\partial \theta} \right\rangle = 2 \left\langle \Psi_\theta | \hat{H} | \frac{\partial \log \Psi_\theta}{\partial \theta} \Psi_\theta \right\rangle \tag{17}$$

$$= 2 \left\langle \Psi_\theta | \hat{E}_{\text{loc}} | \frac{\partial \log \Psi_\theta}{\partial \theta} \Psi_\theta \right\rangle \tag{18}$$

As for the derivative in the second term, some manipulations yield:

$$\frac{\partial}{\partial \theta} \frac{1}{\langle \Psi_\theta | \Psi_\theta \rangle} = -\frac{2}{\langle \Psi_\theta | \Psi_\theta \rangle^2} \left\langle \frac{\partial \Psi_\theta}{\partial \theta} | \Psi_\theta \right\rangle \tag{19}$$

$$= -\frac{2}{\langle \Psi_\theta | \Psi_\theta \rangle^2} \left\langle \frac{\partial \log \Psi_\theta}{\partial \theta} \Psi_\theta | \Psi_\theta \right\rangle \tag{20}$$

Substituting these expressions into (15), we find

$$
\begin{aligned}
\frac{\partial}{\partial\theta}\left[\frac{\langle\Psi_\theta|E_{\mathrm{loc}}|\Psi_\theta\rangle}{\langle\Psi_\theta|\Psi_\theta\rangle}\right] &= \frac{\partial\langle\Psi_\theta|E_{\mathrm{loc}}|\Psi_\theta\rangle}{\partial\theta}\frac{1}{\langle\Psi_\theta|\Psi_\theta\rangle} + \langle\Psi_\theta|E_{\mathrm{loc}}|\Psi_\theta\rangle\frac{\partial}{\partial\theta}\frac{1}{\langle\Psi_\theta|\Psi_\theta\rangle} \\
&= \left[2\left\langle\Psi_\theta|\hat{E}_{\mathrm{loc}}|\frac{\partial\log\Psi_\theta}{\partial\theta}\Psi_\theta\right\rangle\right]\frac{1}{\langle\Psi_\theta|\Psi_\theta\rangle} \\
&\quad - \langle\Psi_\theta|E_{\mathrm{loc}}|\Psi_\theta\rangle\left[\frac{2}{\langle\Psi_\theta|\Psi_\theta\rangle^2}\left\langle\frac{\partial\log\Psi_\theta}{\partial\theta}\Psi_\theta|\Psi_\theta\right\rangle\right] \\
&= 2\frac{\left\langle\Psi_\theta|\hat{E}_{\mathrm{loc}}|\frac{\partial\log\Psi_\theta}{\partial\theta}\Psi_\theta\right\rangle}{\langle\Psi_\theta|\Psi_\theta\rangle} - 2\frac{\langle\Psi_\theta|E_{\mathrm{loc}}|\Psi_\theta\rangle}{\langle\Psi_\theta|\Psi_\theta\rangle}\frac{\left\langle\frac{\partial\log\Psi_\theta}{\partial\theta}\Psi_\theta|\Psi_\theta\right\rangle}{\langle\Psi_\theta|\Psi_\theta\rangle}
\end{aligned}
$$

At first, this might not appear like much of an improvement. Recognizing, however, that every term here can be written as an expectation against $\Psi_\theta^2$, makes clear its usefulness

$$
\nabla E(\theta) = 2\operatorname*{\mathbb{E}}_{x\sim\Psi_\theta^2}\left[E_{\mathrm{loc}}(x)\nabla\log\Psi_\theta(x) - 2\left(\operatorname*{\mathbb{E}}_{y\sim\Psi_\theta^2}[E_{\mathrm{loc}}(y)]\right)\nabla\log\Psi_\theta(x)\right] \tag{21}
$$

In practice, we will estimate (4) using a finite collection of samples drawn from the normalized density $\frac{\Psi_\theta^2}{\langle\Psi_\theta|\Psi_\theta\rangle}$.

$$
\nabla E(\theta) = 2\frac{1}{n}\sum_{i=1}^n\left[E_{\mathrm{loc}}(x_i)\nabla\log\Psi_\theta(x_i) - 2\left(\frac{1}{n}\sum_{i=1}^n[E_{\mathrm{loc}}(x_i)]\right)\nabla\log\Psi_\theta(x_i)\right] \tag{22}
$$

Again, this can thankfully be achieved through standard MHMC techniques without having to estimate the normalization constant.

**Remark: Isn't that estimate biased?**
Mathematically, it is true that since we are reusing the same batch of $x_i$ to estimate both expectations in (21) via (4), there will be bias introduced – the covariance between the two portions of the second term will be non-zero. Vanishing variance of the energy, however, will save us again here, since we can upper bound the covariance using Cauchy-Schwarz as

$$
X = \frac{1}{n}\sum_{i=1}^n[E_{\mathrm{loc}}(x_i)] \qquad Y = \frac{1}{n}\sum_{i=1}^n[\nabla\log\Psi_\theta(x_i)] \qquad \mathrm{cov}(X,Y) \le \sqrt{\mathrm{Var}(X)\mathrm{Var}(Y)} \tag{23}
$$

and we know $\mathrm{Var}(X) \longrightarrow 0$ as our estimate of the ground state improves. So for practical purposes, this correlation is negligible.

### A.3 Proof that the sortlet is once-differatiable

*Proof.* By assumption, $\alpha$ is itself assumed smooth ($C^\infty$), so since $\sigma$ is constant except for points in the set

$$
D = \left\{r \in \mathbb{R}^{3N} : \exists i \in \{0..N\} \text{ such that } \alpha_i(r) = \alpha_{i+1}(r)\right\} \tag{24}
$$

$\Psi_\alpha$ is also smooth on $\mathbb{R}^{3N}\backslash D$. Now for $r \in D$, continuity of $\Psi_\alpha$ at $r$ is straightforward since $\prod_{i=1}^N(\alpha_{i+1}(r) - \alpha_i(r)) = 0$, which exactly cancels the discontinuity created by $\sigma(r)$.

**Once-Differentiability**
As for differentiability, we claim that the derivative is given by

$$
\frac{\partial\Psi_\alpha}{\partial r^i}(x) = \left(\frac{\partial\alpha_{k+1}}{\partial r^i} - \frac{\partial\alpha_k}{\partial r^i}\right)\prod_{j=1,j\ne k}^N(\alpha_{j+1}(r) - \alpha_j(r)) \tag{25}
$$

where $k$ satisfies $\alpha_{k+1} - \alpha_k = 0$. By smoothness, we know this expression is correct on $\mathbb{R}^d\backslash D$, so the claim is that it extends, as one might expect, to $D$ as well. To see this, let's consider two cases:

1. $k$ **is non-unique.** Then it's easy to see that $\frac{\partial \Psi_\alpha}{\partial r^i} = 0$, since the resulting sum from applying the product rule always has a zero in each term, canceling any discontinuity potentially resulting from $\sigma(\pi_\alpha)$. To put this more rigorously, the limit from any direction $\lim_{r' \longrightarrow r} \frac{\partial \Psi_\alpha}{\partial r^i}(r') = 0$.

2. $k$ **is unique**. Handling the case where $k$ is unique requires a little more delicate manipulation. Consider that locally, for any $q \in B_\varepsilon(r)$

$$\psi_\alpha(q) = \sigma(\pi_\alpha)\left(\alpha_{k+1}(q) - \alpha_k(q)\right) \prod_{j=1, j \neq k}^{N} \left(\alpha_{j+1}(q) - \alpha_j(q)\right) \tag{26}$$

$$= \left(\widehat{\alpha}_{k+1}(q) - \widehat{\alpha}_k(q)\right) \prod_{j=1, j \neq k}^{N} \left(\alpha_{j+1}(q) - \alpha_j(q)\right) \tag{27}$$

where $\widehat{\alpha}_k$ and $\widehat{\alpha}_{k+1}$ are the same outputs of $\alpha$, evaluated locally, but ordered according to a fixed $\pi_{\hat{\alpha}}(r)$. The exact $\varepsilon > 0$ can be computed by taking a minimum over all $\epsilon_i$ for $i \neq k$, defined to each satisfy

$$(\alpha_{i+1}(y) - \alpha_i(y)) > 0 : \forall y \in B_{\varepsilon_i}(x) \tag{28}$$

This means that locally, $\Psi_\alpha$ is smooth, since the sorted order of the other $\alpha_j$ does not change inside this neighbourhood.

$\square$

## B  EXPERIMENTAL DETAILS

All results in Figure 3a, Figure 3b and Section 3.1 were produced by modifying the FermiNet codebase to use our sortlet instead of the determinant. For the values in Figure 3b we optimized the wavefunction for 20,000 iterations, but for the others in Section 3.1, we ran each for 100,000 iterations. For the smaller systems, $Li, LiH, Be, H_4$ we used a batch size of 512. For those with more electrons, $B, C, N$ we used between 2048 and 4096. We changed the initialization envelope parameters pursuant to [14], but we found that initializing to the suggested $Z$/row value too unstable with our ansatz, so we used 2 instead for Boron and Carbon. All training runs were completed using 2 A6000 GPUs, and ran for approximately 12 hours on average. The energy values except for $N, CH_4$ in Section 3.1 were computed by averaging over 10,000 separate estimates, with 500 mcmc equilibriation steps in between. For $N, CH4$, due to time constraints, the values were estimated using the last 5000 iterations of training. In both cases, the $3 - \sigma$ confidence interval reflected by the parenthesis $(x)$ around a digit where estimated using the Gaussian formula from the central limit theorem as $3\sigma = 3\frac{\hat{\sigma}}{\sqrt{n}}$. Molecules below $C$ in Section 3.1 used $K = 16$ determinants, but those above used $K = 32$.