# OpenReview forum: "Sorting Out Quantum Monte Carlo"
_ICLR.cc/2024/Conference — Submitted to ICLR 2024_

### Official Review · Reviewer_TwGB · 2023-10-13

**Soundness:** 3 good
**Presentation:** 3 good
**Contribution:** 3 good
**Rating:** 6
**Confidence:** 4

**Summary:**

This work introduces an interesting neural-network architecture that is antisymmetric, for continuous input variables and potentially usable as an alternative to determinants in neural wave functions applications.

**Strengths:**

The paper is well written and describes the idea in good detail. It is also the first reported application of a sort-based idea that achieves chemical accuracy, at least on small systems.

**Weaknesses:**

The ansatz presented is still applied to relatively small systems. Also, I am fundamentally worried that given the discontinuity, the kinetic energy is not well defined (see questions). As a minor point, the discussion could be slightly improved, mentioning for example that second-quantized approaches are also determinant-free and based on sorting fermions (e.g. https://www.nature.com/articles/s41467-020-15724-9 ).

**Questions:**

Besides the small (minor) comment on the presentation/connection to determinant-free second-quantized approaches (see above), I mainly have two more in-depth questions :

1. My understanding of this and related approaches is that the resulting wave function is not continuous in space. How does this affect the evaluation of the kinetic contribution to the local energy and how can the authors be sure that the resulting discontinuity does not affect the boundary conditions of the problem? It is indeed well known that if one has a singularity in the laplacian of the wave function, the energy can even violate the variational bound and be lower than the exact one.

2. One appealing reason why Slater Determinants are used is because the non-interacting (or, better, mean-field) limit is exactly recovered by the Determinant form. Can the authors prove that a sortlet can efficiently recover (maybe even in a supervised way, if an analytical proof is not possible) the Hartree Fock limit?

---

### Official Review · Reviewer_G9wz · 2023-10-17

**Soundness:** 3 good
**Presentation:** 4 excellent
**Contribution:** 1 poor
**Rating:** 3
**Confidence:** 4

**Summary:**

In "Sorting Out Quantum Monte Carlo", the authors propose an antisymmetrizer for neural wavefunction ansätze in quantum Monte Carlo methods based on sorting. Compared to the typical antisymmetrization scheme based on determinants, which scales $O(N^3)$ with the number of electrons $N$, the proposed "sortlet" antisymmetrization has $O(N\log N)$ complexity. The overall complexity of evaluating a sortlet wavefunction is given as $O(N^2\log N)$ due to the $O(N^2)$ complexity of evaluating the $N$ terms $\alpha_i(r)$ used in the construction of a sortlet (Eq. 6). The proposed method is applied to variational quantum Monte Carlo calculations of atoms and small molecules with at most ten electrons and compared to an $O(N^2)$-scaling ansatz based on the Vandermonde determinant and the standard $O(N^3)$ antisymmetrization based on determinants. While the sortlet ansatz is superior to the Vandermonde ansatz in all cases and reaches chemical accuracy for some of the smallest systems, its performance for more complicated systems such as the methane molecule is significantly below the standard determinant-based ansatz.

**Strengths:**

The paper is well written and the proposed method clearly described. The section on variational quantum Monte Carlo is an excellent introduction for readers without a quantum chemistry education and makes the topic very approachable.

**Weaknesses:**

The proposed sortlet ansatz is only tested on very small systems. It performs significantly worse compared to the standard determinant-based ansatz for systems as small as the nitrogen atom (7 electrons). While the authors admit that this is the case ("our results are far from competitive with those of neural network ansatz with full determinant"), I do not agree with their statement that the proof-of-concept described in this paper is sufficient evidence that the sortlet ansatz is a promising direction and that their method could become competitive simply by means of additional investments in software engineering. The main problem I see is that the sortlet ansatz, which requires discontinuous $\alpha$ to be universal, is inherently at odds with using a neural network-based wavefunction, since neural networks are typically quite bad at representing discontinuities. From the results presented in the paper, it is unclear how the sortlet ansatz could be improved in the future to reach competitive results. I think the paper would benefit from additional experiments on larger systems (to assess whether the sortlet ansatz systematically becomes worse with increasing system size) and a detailed analysis *why* the ansatz fails (e.g. for nitrogen).

**Questions:**

1. The authors show that increasing the number of sortlets reduces the error w.r.t. the ground state for boron (Fig. 3a). How do similar curves look for the "failure cases" (e.g. nitrogen)? Is it possible to improve the results by simply adding more sortlets? If not, can the authors identify reasons *why* this is not possible?

2. The $O(N\log N)$ scaling of sorting versus the $O(N^3)$ complexity of computing a determinant is presented as one of the main motivations of the work. Given that the sortlet ansatz currently only works well for small systems, I wonder how large the computational advantage is in practice. I assume that for methods like FermiNet/PsiFormer, the calculation of the determinant is not actually the bottleneck (for small systems). I therefore suggest the authors show timing comparisons for training until chemical accuracy for the different methods (sortlet/Vandermonde/determinant).

3. When looking at the results presented in Fig. 2, in particular the results for the isolated atoms, it seems that the sortlet method starts to fail as soon as there is more than one unpaired electron in a given subshell. I suggest that the authors try to determine why this is the case. An analysis of the reasons why the sortlet ansatz fails would allow to judge whether it is reasonable to expect improvements with additional engineering effort in the future, or whether such an effort is futile. It might also allow insights into how alternative sub-cubic antisymmetrization methods could (or couldn't) be designed.

4. A very minor point about the use of "ansatz": There are several occurences of the word "ansatz" where the plural form "ansätze" would be the gramatically correct form (e.g. "when designing ansatz" should be "when designing ansätze" or "when designing an ansatz"). I suggest to change "ansatz" to the gramatically correct form "ansätze" where appropriate.

---

> ### Author Response · Authors · 2023-11-17
>
> Thanks for your review! So when it comes to universality, it is true that our proof, in its construction, uses a discontinuous $\alpha$ equivariant function. But, as pointed out by another reviewer, the proof given in the original ferminet paper also has the same problem. So if this was a serious issue, like you say “fundamentally at odds” with continuity of neural network used as input, then other neural network wavefunction methods employing a full determinant would have the same problem – and be unable to accurately model the ground state. This is of course not the case for them, and we believe it also isn’t the fundamental issue for us as well.
>
> Rather we think a large part of the performance discrepancy can be explained by design decisions that have all been motivated by their success with determinant based wavefunctions – i.e the use of the KFAC approximation to stochastic-reconfiguration, the choice of MCMC variant, along with the network architecture and hyperparameters, among other things. Additionally, since the pretraining method used by FermiNet / PsiFormer relies on fitting to the mean-field orbitals, we were unable to apply it with our model. Designing alternative pretraining procedures, as well as modifying the hyperparameters was something we experimented with, but due to computational constraints we were unable to sweep extensively over these choices.
> We still maintain that our ability to outperform the other $O(N^2)$ vandermonde ansatz is evidence the sortlet approach has merit. We will also be updating the paper to more clearly show the advantages in that regard.

---

> ### Comment · Reviewer_G9wz · 2023-11-20
>
> I thank the authors for the clarification regarding their universality proof. My original criticism regarding the discontinuity was not formulated clearly enough: My primary concern is not about the proof itself, but about the discontinuity of the local energy, which both Reviewer o8Z6 and Reviewer TwGB seem to share.
>
> Regarding the performance discrepancy: Contrary to the authors, I do not think that it can be explained only by design decisions / network architecture / hyperparameters etc. If this were the case, I think the authors should spend more time improving their model to reach better performance. If significant performance gains were indeed as easy as the authors make them out to be, the current results seem premature for publication.
>
> The authors have not addressed my points regarding the determinant not being the computational bottleneck for the systems studied and identifying reasons for the failure of their method for specific systems (boron, nitrogen), which I think could be very insightful. I therefore keep my score as is.

---

### Official Review · Reviewer_1VSk · 2023-10-27

**Soundness:** 2 fair
**Presentation:** 2 fair
**Contribution:** 3 good
**Rating:** 5
**Confidence:** 3

**Summary:**

The paper considers an important challenge in Quantum Chemistry, namely calculating the ground-state energy and its corresponding eigenfunction for gas-phase molecule. Recently, it was introduced to use neural networks as wavefunction parametrization in combination with Quantum Monte Carlo to solve this problem. A common issue is the scaling of the so-called Slater determinant to account for the anti-symmetry of the wavefunction. Evaluation of the Slater determinant scales cubically with the number of particles and therefore accounts for a significant part of the computational cost. The following paper proposes a new technique for anti-symmetrization (removing the necessity of evaluating the Slater determinant) with an improved scaling of O(N log N) with N being the number of particles. They show experimentally that for systems up to Bohr that they can reproduce state-of-the-art results and give a proof for weak universality of their ansatz.

**Strengths:**

The following paper considers a challenging topic of removing the dependency on the Slater determinant and by that trying to improve the scaling of recently proposed deep-learning-based wavefunction ansätze. Beside the anti-symmetry constrain of the wavefunction another complexity of the problem at hand is the need for highly accurate solutions. Normally, DL-VMC tries to recover the last 1-2mHa of the total energy.

Therefore, the following paper has in my opinion three key strength:

-	Novelty: The paper introduces to the best of my knowledge a novel idea to reduce the scaling of the anti-symmetrization process. The idea looks promising, and it is able to recover state-of-the-art results, although only for small systems.
-	The paper motivates the underlying problem and therefore introduces the topic to a broader community. (At certain parts I would have preferred a more detailed motivation, see weaknesses)
-	The proposed approach is underlined with a theoretical finding of the potential universality. Additionally, the paper discusses (in detail) their approach in comparison to other methods with similar scaling and motivates its potential strength.

**Weaknesses:**

One concern I have is regarding the claimed scaling of O(N log N) in the abstract. As stated in Sec. 3.1. it is more in the realm of O(N^2 log N) if not worse (see Fig. 3a).
A more detailed discussion about the need of expansions (K) would help improve the paper. With the current results it is difficult for me to assess if the scaling is actually as proposed.

Although the paper motivates the problem and I am fine with the general structure of the paper, I have problems with the notation and explanation of certain concepts:

**Section 1.3. “Where did the determinants come from anyway?”**

Anti-symmetry is introduced in eq. (AS) but it is quite difficult to understand the eq. because:

-	I couldn't find a definition of M.
-	What does the up and down arrow represent?
-	The positions of the electrons r are not introduced the first time they are used.
-	The anti-sym. (at least in the eq.) is only defined for spin-down electrons (or am I missing something?).

**Section 3.1 “The Sortlet ansatz”**

The author might consider explaining their idea in more detail:

-	In case of K>1 you are mapping the $\alpha$ functions to $\mathbb{R}^{N \times K}$, is this correct?
-	Eq. 7 you sum over K with $\alpha_i$, I am assuming the index i is not the same as in the introduction of the ansatz (eq. 6 and following bullet points)?
-	The exponential with factor $\gamma$ is to the best of my knowledge also new compared to prev. introduced wavefunction ansätze, where you would have an exponential envelope over the electron-nuclei distances. Could the author explain their design choices in more detail?
-	In the complexity paragraph, they state $\alpha$ can be evaluated in $O(N^2)$. A short sentence that this is related to the computation of electron-electron interactions would give at least an intuition why this might be true. In my opinion this is especially difficult to understand for a reader who doesn’t know the PauliNet, FermiNet & PsiFormer papers in detail.

I have additional (minor) comments regarding the notation (see “additional comments”).
I want to stress I don’t think the paper needs to describe the whole field of DL-VMC but maybe the authors want to revise certain parts for better readability to reach a broader community.

With the current weaknesses I am hesitant to recommend the paper to be accepted at ICLR. My decision is connected to the questions (see “Questions”) below because I am not convinced regarding the scaling, or in other words the number of expansions (K) needed for accurate results and it is difficult to assess with the given result section.

**Minor additional comments:**

-	The paragraph regarding the discussion of the different neural network architectures in Sec. 1.3. (around “…which is allowed non-symmetric…”) might be difficult to understand for a reader not familiar in detail with the work of FermiNet & PauliNet.
The authors might consider adding additional details to better explain their argumentation (maybe in the SI). This is just a recommendation. I understand that it is not the task of the authors to explain the whole field of DL-VMC, but since the proposed ansatz requires quite a deep understanding of the topic it might be beneficial to add additional context.
-	Sec. 2.4 “Remark: Isn’t that estimate biased”: The authors write “the covariance between the two expectation in (4) via (23)”, do the author maybe mean (22) via (4) and not (23) ((4) and (23) seems to me to be the same equation)?
-	In the proof of Prop. 4. We have n and N, I assume it should be both N?
-	In the Hamiltonian you write $R_I$ but in the notation block $R_i$.

**Questions:**

- Can the author elaborate more on the scaling of their method?
  - How large do you expect K to be? In Fig. 3a. it seems to be way more than linear. But since it is only one experiment it is difficult to assess.
- Do you have an intuition why your results are 0.5 Ha away from the FermiNet results for only a slightly larger system (compared to Bohr) for Nitrogen? You mentioned engineering problems, but this is difficult to assess without having a chance to look at the code (hopefully I didn’t miss a reference to the code).

---

> ### Author Response · Authors · 2023-11-18
> **Clarification on notation issues**
>
> Notation:
>
> Thank you for helping us improve the readability of our paper by pointing out notation details. To address each:
>
> - I couldn't find a definition of M.
>
> Ah yes, M is just the number of spin up electrons. We use the notation $N^{\uparrow}, N^{\downarrow}$ in the rest of the paper, and will make the intro consistent with that choice.
>
>  - What does the up and down arrow represent?
>
> The up and down arrows denote the spin of the particle. The function is only antisymmetric with respect to permutations of particles with the same spin. We will make this clearer.
>
> - The positions of the electrons r are not introduced the first time they are used.
> Thank you for pointing this out, we will fix this!
>
> - The anti-sym. (at least in the eq.) is only defined for spin-down electrons (or am I missing something?).
>
> We can see that this can cause confusion; we will make clear that the anti symmetry needs to be enforced for spin up and spin down.
>
> - In case of K>1 you are mapping the  functions to, is this correct?
>
> That is correct, we all $\alpha$ terms  are the output to from a single neural network backbone
>
> - Eq. 7 you sum over K with, I am assuming the index i is not the same as in the introduction of the ansatz (eq. 6 and following bullet points)?
>
> You are correct, they are different indices, we will clarify this
>
> - The exponential with factor is to the best of my knowledge also new compared to prev. introduced wavefunction ansätze, where you would have an exponential envelope over the electron-nuclei distances. Could the author explain their design choices in more detail?
>
> Ah yes this is also a typo, it was written for the case of a single atom. Of course, for molecular systems we have an exponential envelope over the electron nuclei distances, just like the other ansatze.
>
> - In the complexity paragraph, they state can be evaluated in. A short sentence that this is related to the computation of electron-electron interactions would give at least an intuition why this might be true. In my opinion this is especially difficult to understand for a reader who doesn’t know the PauliNet, FermiNet & PsiFormer papers in detail.
>
> Thank you, we will mention that the $O(N^2)$ complexity stems from the electron-electron interactions.

---

> > ### Comment · Reviewer_1VSk · 2023-12-02
> >
> > I appreciate the effort the authors put into the rebuttal. I am hesitant to raise my score and will leave it as is for now.
> > As the other reviewers pointed out, the discontinuity in the local energy is an open question. Additionally, the experimental results are not convincing, and the authors didn’t discuss this point during the rebuttal.

---

### Official Review · Reviewer_o8Z6 · 2023-10-30

**Soundness:** 2 fair
**Presentation:** 2 fair
**Contribution:** 2 fair
**Rating:** 3
**Confidence:** 5

**Summary:**

This paper proposed a method based on sorting operations to achieve the anti-symmetry property of the NN-VMC ansatz. Compared with the original determinant ansatz, the proposed algorithm has a favorable computational complexity. The authors conduct experiments on small molecule systems, showing the proposed ansatz has a comparable performance as the baseline methods.

**Strengths:**

It is interesting to design an ansatz with favorable computational complexity.

**Weaknesses:**

1. To the reviewer’s best knowledge, the sorting algorithm in 3-dimensional space (more concretely, any spaces where the dimension is larger than one) is discontinuous. Thus, the proposed ansatz is discontinuous. When calculating the kinetic energy term, it is hard for the mcmc walkers to handle the energy near the discontinuous surface. As a result, the conventional energy calculation method in NN-VMC will lead to a non-variational energy result, which means the energy of the proposed method cannot directly compare with the other NN-VMC methods.
 In section 1.2, the authors claim that some previous works ‘allude’ to a non-smooth ansatz. However, the corresponding ansatzes are smooth. The discontinuous property is just an assumption of the theoretical analysis. Therefore, those previous works do not suggest using a discontinuous ansatz. The authors must discuss more about the influence of the discontinuous property in the paper.

2. The computational cost of determinants is very small for the systems within 10 electrons. Thus, the acceleration achieved through the proposed ansatz is relatively small in those systems. To demonstrate the effectiveness of the proposed ansatz, the authors should study larger systems.

**Questions:**

The questions are listed in the weakness

---

> ### Author Response · Authors · 2023-11-14
> **Response to question 1**
>
> We appreciate your effort in reviewing our paper! It seems, however, that there has been some confusion over the design of our proposed ansatz. While our universality proof was based on allowing a discontinuous input function to the sortlet layer –  and as you note, the same concession was made to prove a type of universality in the appendix of the original FermiNet article –  in both cases, this is only relevant to the proof. As we state in Proposition 3 (proof of which can be found in the appendix), our construction is continuous and once differentiable, which is sufficient for the kinetic energy operator to be well defined (by application of the standard Green’s identity $\langle \psi | \Delta \psi \rangle = -\langle \nabla \psi | \nabla \psi \rangle$ )
>
> If you have a specific concern about the proof of proposition 3, we’d be more than happy to clarify any details!

---

> > ### Comment · Reviewer_o8Z6 · 2023-11-16
> >
> > Thank you for your clarification and sorry for the misunderstanding. However, the author still has concerns about the non-smooth property of the proposed ansatz. The Schrodinger equation ensures that $\hat{H}\psi=E\psi$. Thus, if the proposed ansatz is just once differentiable, i.e., has a discontinuous local energy, it can not satisfy the Schrodinger equation. The reviewer hopes the author could discuss this problem.
> >
> > Minor: The index in the appendix eq.(25) may need more clarification or correction, e.g., the index of $\alpha$ in the first bracket should be $k$ rather than $i$.

---

> > > ### Author Response · Authors · 2023-11-19
> > >
> > > No problem. So, it's important to keep in mind that even for the exact physical solution, it's known that the Kato cusp conditions enforce a discontinuity in the gradient of the wavefunction (see Kato, 1958). Therefore, the Schrodinger equation $\hat H \psi = E\psi$ is already known to not be satisfied everywhere in the mathematical sense. By virtue of employing the jastrow factor $J_\beta$, like other neural wavefunctions, we also have deliberately created a cusp at points where the potential $V$ (from the Born-Oppenheimer approximation) has infinite energy. These are used precisely to satisfy the Kato conditions.
> > >
> > > Since the total kinetic energy can be redefined in terms of only the gradient (again using the Green's identity), we did not include a full proof that our wavefunction is twice differentiable, though it is true. The proof is slightly more complicated, but follows directly from extending the logic used to prove once-differentiability. We will update the text to include it, since it seems to be causing some confusion.

---

> > > > ### Comment · Reviewer_TwGB · 2023-11-19
> > > > **Discontinous local energy**
> > > >
> > > > >  it's important to keep in mind that even for the exact physical solution, it's known that the Kato cusp conditions enforce a discontinuity in the gradient of the wavefunction (see Kato, 1958). Therefore, the Schrodinger equation
> > > >  is already known to not be satisfied everywhere in the mathematical sense.
> > > >
> > > > I think the other reviewer was referring to the discontinuity of the local energy E_loc(R)=<R|H|Psi>/<R|Psi> . The exact wave function does not have any discontinuity in the local energy (which is strictly a constant as a function of R, for the exact solution) and the Kato cusp condition is indeed used to enforce that the local energy does not diverge at any point in space.
> > > >
> > > > This issue of discontinuity is also what I meant in my comment. In general, I agree that there can/should be cusps, but these should be such that they satisfy the boundary conditions of the problem (either on the nodes, or on the boundaries of the box, for a periodic system, for example).

---

> > > ### Author Response · Authors · 2023-11-21
> > >
> > > So to respond again to this (and the other reviewers comment), we have updated the appendix / paper with a proof that the sortlet ansatz is continuously second differentiable. This should clarify any issues around the validity of the ansatz w/r/t the local energy being defined.
> > >
> > > We have also fixed the typo in eq (25).

---

> > > > ### Comment · Reviewer_o8Z6 · 2023-11-22
> > > >
> > > > Thank you for your effort in giving that proof. However, the reviewer has found a counter-example so there may exist some mistakes in that proof.
> > > >
> > > > Let's consider that $\alpha(x)=x$. Assume that the input space is one-dimensional and there only exist 3 particles. In the region where $x_1>x_2>x_3$, the proposed ansatz gives that $\psi({\bf x})=(x_1-x_2)(x_2-x_3)$. Thus, in this region, $\partial^2_{x_2}{\psi(\{\bf x})}=-1$. In the region where $x_2>x_1>x_3$, the proposed ansatz gives that $\psi({\bf x})=-(x_2-x_1)(x_1-x_3)$ and $\partial^2_{x_2}{\psi(\{\bf x})}=0$. Therefore, the proposed ansatz's second-order derivative is not continuous. The author may recheck their proof or add more property to $\alpha$ to ensure the continuous.
> > > >
> > > > Besides the discontinuous property, the weakness of system size has not been addressed by the authors. The reviewer would like to point out that this problem is also important. The authors may also pay attention to discuss about this problem.

---

> > > > > ### Author Response · Authors · 2023-11-23
> > > > >
> > > > > Thanks for checking out the argument, but this in itself isn't a counterexample. For 3 particles, the sortlet has 3 terms, so your first expression for $\psi$ should read $\psi = (x_1 - x_2) (x_2 - x_3) (x_3 - x_1)$ and so on.
> > > > >
> > > > > However, generalizing this to a system of 4 particles does seem to produce a counterexample. The issue with the proof is that the expression we calculate for the second derivative is wrong to begin with (31), there are terms missing.
> > > > >
> > > > > If we take a step back for a moment, it's important to note that we don't need our wavefunction to be twice differentiable to remain variational - that is, for our results to upper bound the energy of the system. So this lack of smoothness does not make our approach invalid, although it might play a part in explaining why our results do not approach those of the other determinant based neural ansatz, since it makes the inductive bias of our wavefunction worse.

---

> > > > > > ### Comment · Reviewer_o8Z6 · 2023-11-23
> > > > > >
> > > > > > Thank you for your correction and clarification. The total energy of the ansatz is well-defined if the ansatz is once-differentiable. Although the reviewer still has concerns about evaluating the discontinuous local energy through the standard MCMC method, it is not the main reason for the reviewer to reject the paper.
> > > > > >
> > > > > > However, the reviewer can not increase the score because another important problem has not been addressed by the authors. The systems studied in this paper are so small that the efficiency improvement of the proposed ansatz is almost negligible, and the total energy of the proposed ansatz is significantly higher than the original ansatz. Combined with the discontinuous property discussed before, the reviewer can not give a higher score for the current version.

---

### Author Response · Authors · 2023-11-23

We thank the reviewers for their time and effort in helping us revise our manuscript. As the rebuttal period draws to a close, we'd like to summarize some of our responses in this top level comment, as well as highlight some changes we have made to the paper.

We would like to reiterate that since we are in an entirely different complexity class than determinant based ansatz, direct comparisons are arguably not appropriate. Showing that the sorting approach can work at all, we believe, is a surprising result in itself, and offers an improvement over the vandermonde ansatz, which is a more equitable comparison since it is at least in the same complexity class. To that end, we have added two comparison graphs to the final revision of the paper, further highlighting how our sortlet ansatz significantly outperforms its vandermonde counterpart.

As for the smoothness concerns some of the reviewers have raised, we maintain here that these are not sufficient to undermine any of the results we have presented. Since our construction is once differentiable and satisfies the decay, and of course, the antisymmetry conditions, it is an admissible wavefunction for the variational optimization. Further studying how the cusps in the first derivative affect the energy, however, might lead to deeper insights in how to close the gap between our ansatz and chemical accuracy on larger systems.

The goal of research paper should be to give a proof of concept of a new idea. By introducing a novel ansatz with favourable scaling that is, somewhat surprisingly, smooth enough for use in QMC, then demonstrating it performs acceptably on a number of small systems, we believe to have done as much.

---

### Meta-Review · Area_Chair_oz2H · 2023-12-07

**Metareview:**

The paper presents an interesting idea for using neural networks as ansatzes for variational quantum monte carlo calculations with a very favorable asymptotic complexity. Rather than using a determinant, which scales cubically, to construct an antisymmetric ansatz, the input vectors are sorted along some axis, which scales as O(nlog(n)). In practice, this is superior to DeepWF, another subcubic neural network ansatz for electronic structure, but inferior to determinant-based ansatzes. All reviewers raised concerns that the ansatz was not smooth. While the authors explain that the ansatz is once-differentiable everywhere, it is not *twice* differentiable at the nonsmooth points, which may cause pathological behavior similar to cusps. Consistent with the judgment of the reviewers, I recommend that the paper be rejected - while the results are clearly better than DeepWF, DeepWF itself was primarily notable for being the first neural network ansatz, not because it was particularly good.

I do however believe the idea is promising and could be quite impactful if it were developed further. A few recommendations for the authors:

- There is already a function that defines an ordering on particle locations *and* is smooth - the determinant. The determinant is usually computed numerically via a pivoted LU decomposition, and the pivot defines a unique ordering for the particle locations. Perhaps a careful consideration of how the pivoted LU decompositon works could inspire a similar smooth sorting algorithm which has subcubic scaling.

- Dario Bressanini has done significant work on the nodal surface of the beryllium atom, showing it has a simple topological structure that cannot be captured by a single Slater determinant (e.g. PRB 86, 115120 (2012)). This is even true for the generalized Slater determinants used by neural network ansatzes. The true ground state has only two "pockets" - one positive, one negative - while a Slater determinant will find 4 regions which all connect at a single point. This might be a good, simple model system for analyzing subcubic ansatzes that could provide useful insight.

I hope this helps, and wish them better luck in the future.

**Justification For Why Not Higher Score:**

I don't understand why this field is included. A good metareview should answer this question already. The review process is already demanding enough, we do not need to create extra work for everyone.

**Justification For Why Not Lower Score:**

I don't understand why this field is included. A good metareview should answer this question already. The review process is already demanding enough, we do not need to create extra work for everyone.

---

### Decision · Program_Chairs · 2024-01-16

Reject